# Comparative Metabolomics Analysis Reveals Sterols and Sphingolipids Play a Role in Cotton Fiber Cell Initiation

**DOI:** 10.3390/ijms222111438

**Published:** 2021-10-23

**Authors:** Qiaoling Wang, Qian Meng, Fan Xu, Qian Chen, Caixia Ma, Li Huang, Guiming Li, Ming Luo

**Affiliations:** 1Key Laboratory of Biotechnology and Crop Quality Improvement, Ministry of Agriculture/Biotechnology Research Center, Southwest University, Chongqing 400716, China; wql19980513@163.com (Q.W.); mqhongbin@foxmail.com (Q.M.); xufanfeiren@163.com (F.X.); chenqiansuaige@163.com (Q.C.); mcx2116@email.swu.edu.cn (C.M.); hl19970914@email.swu.edu.cn (L.H.); lgm5683@163.com (G.L.); 2Key Laboratory of Horticulture Science for Southern Mountains Regions of Ministry of Education, College of Horticulture and Landscape Architecture, Southwest University, Chongqing 400716, China; 3Academy of Agricultural Sciences of Southwest University, State Cultivation Base of Crop Stress Biology for Southern Mountainous Land of Southwest University, Chongqing 400716, China

**Keywords:** sphingolipids, sphingolipid metabolism, sterol, cotton, fiber cell, initiation, differentiation

## Abstract

Cotton fiber is a seed trichome that protrudes from the outer epidermis of cotton ovule on the day of anthesis (0 day past anthesis, 0 DPA). The initial number and timing of fiber cells are closely related to fiber yield and quality. However, the mechanism underlying fiber initiation is still unclear. Here, we detected and compared the contents and compositions of sphingolipids and sterols in 0 DPA ovules of *Xuzhou142 lintless-fuzzless mutants* (*Xufl*) and *Xinxiangxiaoji lintless-fuzzless mutants* (*Xinfl*) and upland cotton wild-type Xuzhou142 (XuFL). Nine classes of sphingolipids and sixty-six sphingolipid molecular species were detected in wild-type and mutants. Compared with the wild type, the contents of Sphingosine-1-phosphate (S1P), Sphingosine (Sph), Glucosylceramide (GluCer), and Glycosyl-inositol-phospho-ceramides (GIPC) were decreased in the mutants, while the contents of Ceramide (Cer) were increased. Detail, the contents of two Cer molecular species, d18:1/22:0 and d18:1/24:0, and two Phyto-Cer molecular species, t18:0/22:0 and t18:0/h22:1 were significantly increased, while the contents of all GluCer and GIPC molecular species were decreased. Consistent with this result, the expression levels of seven genes involved in GluCer and GIPC synthesis were decreased in the mutants. Furthermore, exogenous application of a specific inhibitor of GluCer synthase, PDMP (1-phenyl-2-decanoylamino-3-morpholino-1-propanol), in ovule culture system, significantly inhibited the initiation of cotton fiber cells. In addition, five sterols and four sterol esters were detected in wild-type and mutant ovules. Compared with the wild type, the contents of total sterol were not significantly changed. While the contents of stigmasterol and campesterol were significantly increased, the contents of cholesterol were significantly decreased, and the contents of total sterol esters were significantly increased. In particular, the contents of campesterol esters and stigmasterol esters increased significantly in the two mutants. Consistently, the expression levels of some sterol synthase genes and sterol ester synthase genes were also changed in the two mutants. These results suggested that sphingolipids and sterols might have some roles in the initiation of fiber cells. Our results provided a novel insight into the regulatory mechanism of fiber cell initiation.

## 1. Introduction

Cotton fiber is an extremely elongated single cell of seed epidermis and an important raw material for the textile industry. The growth and development of fibers can be divided into four distinct and overlapping periods: initiation, elongation (primary cell wall formation), secondary cell wall (SCW) deposition, and maturation [1,2,3]. Based on the length of mature fibers, cotton fibers are divided into lint and fuzz. Lint fiber begin to protrude from 0 DPA (Day Post Anthesis) to 3 DPA and a large number of spherical bulges can be observed on the ovule surface at 0 DPA. Then, fiber cells enter the elongation stage. Yield and quality of mature cotton fibers are closely correlated with the timing of fiber initiation and the number of initiated fibers [4]. About 30% of the seed epidermal cells differentiate into spinnable fibers [5]. Therefore, elucidation of the mechanisms of fiber initiation is valuable to the cotton industry. In the past two decades, many genes involved in fiber cell initiation have been documented. Loguerico et al. identified six MYB-related genes potentially involved in the differentiation and development of cotton seed trichomes by a PCR-based strategy. Three of them (GhMYB-1, -2, and -3) transcripts were found in all tissue-types examined and were relatively more abundant than the others (GhMYB-4, -5, and -6), which showed distinct, tissue-specific expression patterns [6]. R2R3 MYB transcription factor GhMYB25-like has an important role in fiber initiation. Silencing of *GhMYB25-like* in cotton resulted in fibreless seeds, similar to the *lintless**-fuzzless mutant XU142fl* [7]. MYB109 and MYB2 transcription factors were expressed in the stage of fiber initiation [8]. GaMYB2 transcription factor can restore *Arabidopsis thaliana* trichome mutant and activate the expression of *R22-like 1* (*RDL1*) gene in fiber primordium [9]. Furthermore, *RDL* genes and the genes related to cell structure, long-chain fatty acid biosynthesis, and sterol biosynthesis have been identified as deletion or reduction in cotton fibreless mutants [10]. Two WD-repeat genes, *GhTTG1* and *GhTTG3* could restore the trichome development of *Arabidopsis thaliana ttg1* mutant [11]. By a cDNA microarray, 57 and 15 genes were identified as highly expressed in the initial fiber cells and epidermal cells of the cotton ovules, respectively [12]. IV HD-Zip family transcription factor *GhHOX1* can rescue trichome formation of *gl2-2* mutant in *Arabidopsis thaliana*, which may be involved in cotton fiber initiation [13]. Comparative transcriptome analysis of WT and its near isogenic *fl* mutant (MCU5) revealed that the genes involved in calcium and phytohormone mediated signal transduction pathways, biosynthesis of auxin and ethylene, and stress responsive transcription factors were down-regulated in *fl* mutant [14]. bHLH transcription factor *GhDEL65* can rescue trichome initiation in *Arabidopsis thaliana gl3egl3* mutant and participate in regulating cotton fiber elongation [15]. Hu et al. have identified 645 lncRNAs that expressed preferentially in the fibreless mutant Xu142 *fl* and 651 lncRNAs that expressed preferentially in the fiber-attached lines. Of which, three lncRNAs were further confirmed that played an important role in fiber development by a virus-induced gene silencing (VIGS) system [16]. Wang et al. reported that the expression of genes involved in carboxylic acid metabolism, small-molecule metabolic processes, hormone regulation and lipid metabolism were significantly enhanced in wild-type ovules when compared with *lintless-fuzzless mutant*. The *GhPAS2* (3-hydroxyacyl-CoA dehydratase) involved in VLCFA biosynthesis accumulated at anthesis in wild-type ovules [17].

Among the progress that has been made, the main studies focused on the screening of differentially expressed genes and their functional identification. There are few studies on the roles of biochemical metabolites in the initiation of fiber cells. It is reported that H_2_O_2_ played certain roles in fiber initiation and might be an upstream signal molecule regulating the expression of *GhMYB25* and *GhEXP1* [18,19]. Taliercio and Boykin reported that the expression of “membrane” related genes was significantly higher in wildtype than in the fibreless mutant at fiber initiation stage. Staining ovules with fluorescent dyes confirmed that the endoplasmic reticulum (ER) of fibers increased in 0 DPA, lasted for 3 DPA, and was missing in the fibreless mutant. It is suggested that membrane might play an important role in fiber initiation [20].

The initiation of cotton fiber is also a polar expansion of cells. Dynamics of membrane plays an essential role in the establishment of cell polarity [21]. Sphingolipids and sterols are two important components of cell membrane, which are mainly concentrated in the functional region of membrane-lipid raft [22,23]. Sphingolipids are a structurally diverse group of molecules based on long-chain sphingoid bases [24]. The sphingolipids molecule consists of three main components: the long chain base (LCB) of sphingosine, the long chain fatty acids (LCFA) or the very long chain fatty acids (VLCFA), and the polar head group. The two chains are linked to the polar head by amide bonds to form ceramide (Cer), which is the backbone of complex sphingolipids [25,26,27]. Sterols are isoprenoid-derived molecules. In higher plants, a great number of phytosterols are synthesized, among which sitosterol, campesterol and stigmasterol are the predominant forms [23,28].

Previous studies have revealed that both sphingolipids and sterols play important roles in the establishment of cell polarity. Glycosphingolipids (GSLs) can provide a molecular platform for recruitment signal transductor. Sphingolipids and sterols form membrane microdomains (lipid rafts), which are important in the generation of polarized membrane domains and the sorting and transportation of intracellular proteins [21]. Liu et al. reported that lipid microdomains are involved in NADPH oxidase (NOX)-dependent reactive oxygen species (ROS) signal transduction in the polar growth of spruce pollen tubes [29]. Markham et al. reported that sphingolipids with very long acyl chains define a trafficking pathway with specific endomembrane compartments and polar auxin transport protein cargoes [30]. By proteomic and genomic analysis, Glycosylphosphatidylinositol (GPI)-anchored proteins were identified from *Arabidopsis thaliana*, their homologues are associated with sterol rich lipid rafts in animal cells [31]. Further studies showed that one of the GPI-anchored proteins COBRA (COB) was asymmetrically distributed in polarized cells, which was required for anisotropic expansion of cells. The cob mutant leads to loss of polarity in root cell expansion of *Arabidopsis thaliana*. The COB protein contains GPI-anchored sequence, which anchored to the extracellular surface of the plasma membrane by GPI. The COB protein was mainly distributed in the longitudinal side of root cells in the rapid elongation zone and its expression level was significantly up-regulated in cells entering the rapid elongation region [32].

Willemsen and colleagues identified the orc mutant, which is a loss-of-function mutant for the *Arabidopsis thaliana STEROL METHYLTRANSFERASE1* (*SMT1*), a C-24 sterolmethyl transferase. This mutant accumulates cholesterol and campesterol rather than sitosterol, the major plant sterol. In wildtype, PIN1 is basally localized and PIN3 localizes uniformly at the plasma membrane, while the proteins harboring a lateral localization in root cells in the orc mutant, indicating that the polar localization of PIN1 and PIN3 was disrupted in orc mutant [33]. The other loss-of-function mutant for the *Arabidopsis thaliana CYCLOPROPYLSTEROL ISOMERASE1-1* (*CPI1-1*) was identified. Compared with wild type, the sterol composition strongly altered in cpi1-1 mutant that displays a severe dwarf phenotype and a strong defect in the root gravitropism response. Consistently, PIN2 localization was impacted after cytokinesis [34]. These results indicated that sphingolipids and sterols are required for the establishment of cell polarity.

As many previous studies mentioned, *fuzzless-lintless* mutant is a good material for studying fiber cell initiation. *Xuzhou 142 fuzzless-lintless* mutant (*Xufl*) was isolated from the cotton cultivar G. hirsutum cv. Xuzhou142 [35]. The *Xufl* plants show no phenotypic difference from the wild type (XuFL), except that the *Xufl* seeds are glabrous [36]. Genetic analysis indicated that the *Xufl* mutant was derived from a single recessive mutation from the wild type of Xuzhou142 (XuFL) [35]. In the last two decades, The XuFL and its *fuzzless-lintless* mutant (*Xufl*) were widely used in studying the early development of cotton fiber cell [16,19,36,37]. *Xinxiang Xiaoji fuzzless-lintless* mutant (*Xinfl*) was found in the cotton field of Xiaoji Town, Xinxiang County, Henan Province, China, in 1991. Genetic analysis showed that its wild type may be Yumian 4#. There was on phenotypic difference, except for fuzzless-lintless phenotype between the *Xinfl* plants and cotton cultivar *G. hirsutum* cv. Xuzhou142 or Yumian 4#. Moreover, Wang et al. have reported that the index of genetic identity was 0.9 between the *Xinfl* mutant and the *Xufl* mutant indicating the two *fuzzless-lintless* mutants had great genetic similarity [38]. Therefore, the two mutants often used in the study of revealing the genes and regulatory mechanism related to cotton fiber cell initiation [39].

In order to clarify the role of sphingolipids and sterols in the initiation of cotton fiber cells, the contents and compositions of sphingolipids and sterols in the 0-DPA ovules of upland cotton XuFL and two lintless-fuzzless mutants, *Xufl* and *Xinfl* were detected by UHPLC–MS/MS. The differences of sterols and sphingolipids and the expression of related genes in the three samples were analyzed. Furthermore, exogenous application of a gluceramide synthesis inhibitor, PDMP (1-phenyl-2-decanoylamino-3-morpholino-1-propanol), in ovule culture system obviously inhibited the initiation of cotton fiber, implying that the change of sphingolipids may be an important reason for the suppression of fiber initiation in two lintless-fuzzless mutants. These results suggest that sphingolipids and sterols may have some roles in the initiation of cotton fiber cells, which provides a novel insight for the further study of the regulatory mechanism of cotton fiber differentiation and initiation.

## 2. Results

### 2.1. The Profile and Content of Various Sphingolipids in 0-DPA Ovules of Upland Cotton

The fiber cell is formed by the protrusion of some epidermal cells of the ovule. On the day of anthesis (0-DPA), the protruded fiber cells can be seen on ovule surface of normal upland cotton. However, there is no fiber cell protrusion in Xuzhou142 *lintless-fuzzless mutant* (*Xufl*), and *Xinxiangxiaoji lintless-fuzzless mutant* (*Xinfl*) (Figure 1A), which also have no fiber on the coat of mature seed (Figure 1B). In order to understand the difference of sphingolipid compositions and contents between upland cotton wild-types and lintless-fuzzless mutants during the initiation of fiber cells, we detected the sphingolipid compositions in 0-DPA ovules of Xuzhou142 wild-type (XuFL), *Xufl*, and *Xinfl* by UHPLC–MS/MS (The original data were in Appendix A). The results showed that 9 classes of sphingolipids and 66 sphingolipid molecular species were detected in 0-DPA ovules (with or without fiber cells), including PhytoSphingosine-1-phosphate (t-S1P), PhytoSphingosine (PhytoSph), Sphingosine (Sph), Ceramide (Cer), PhytoCeramide (PhytoCer), PhytoCeramides with hydroxylated fatty acyls (PhytoCer-OHFA), GlucosylCeramides(GluCer), Phyto-GlucosylCeramides (Phyto-GluCer), and Glycosyl-Inositol-Phospho-Ceramides (GIPC). Their number of molecular species is 2, 2, 1, 12, 13, 17, 9, 8, and 2, respectively (Figure 2A). The molecular species of complex sphingolipids are far more than that of simple sphingolipids, particularly, the molecular species of Cer and GluCer accounted for 64% and 26% of the total, respectively (Figure 2A). The four sphingolipids with the highest content were PhytoSph, PhytoCer-OHFA, PhytoCer and Phyto-GluCer (Figure 2B), while the four sphingolipids with the lowest content were t-S1P, Sph, GluCer, and GIPC (Figure 2B). Compared with wild-type XuFL, the contents of PhytoSph, Sph, GluCer and GIPC were decreased, while the contents of Cer were increased in the two mutants (Figure 2B). These results showed that the predominant sphingolipids in 0-DPA ovules were sphingolipids with tri-hydroxyl LCB and some classes of sphingolipids altered in content between wild-types and mutants.

### 2.2. The Composition and Content of Sterols in 0-DPA Ovules

Sphingolipid and sterols are the components of membrane lipid raft, and also are interdependent and influence each other. Therefore, we also detected the sterol contents and compositions in 0-DPA ovules of Xuzhou 142 wild type and two mutants by UHPLC–MS/MS (The original data were in Appendix A). The results showed that 5 sterols (cholesterol, campesterol, sitosterol, stigmasterol, and stigmasterol) and 4 steryl esters (campesterol esters, sitosterol esters, stigmasterol esters and stigmastanerol esters) were detected in 0-DPA ovules (Figure 3A). In 0-DPA ovules, the content of free sterols is higher than that of conjugated sterols. The order of free sterol content from high to low is sitosterol, campesterol, stigmasterol, stigmastanol, and cholesterol. The order of steryl ester content from high to low is sitosterol esters, campesterol esters, stigmasterol esters, and stigmastanerol esters (Figure 3A). Compared with wild type, the total amount of steryl esters and the contents of four steryl esters were increased in the two mutants although the increase level of some substances was not significant. The contents of sitosterol and stigmastanol had no difference between wild type and mutants. The content of campesterol and stigmasterol was significantly higher in mutant ovules than in wild-type ovules, increased by 15.7% and 31.2%, respectively. Compared with wild type, the cholesterol content in mutant 0-DPA ovules was decreased by 18% (Figure 3A). The contents of campesterol esters and stigmasterol esters in mutant ovules were significantly increased (79% and 110%, respectively). Given that the ratio of campesterol to sitosterol (C/S) plays important roles in plant growth and development [40,41], we further analyzed the C/S ratio. The results showed that the C/S value was significantly higher in the mutants than in wild type (Figure 3B). These results indicated that the two free sterols and corresponding steryl esters increased in the 0-DPA ovules of two fiber initiation mutants.

### 2.3. The Difference of Simple Sphingolipids between Wild-Type and Mutants

Simple sphingolipids mainly include four molecular species, namely sphingosine (Sph), ceramides (Cer) or phytoceramides, and their phosphorylated derivatives sphingosine-1-phosphate (S1P) and ceramide-1-phosphate (CerP). Complex sphingolipids include two molecular species, GCS and GIPC [42]. In opposition to simple sphingolipids, complex sphingolipids have complex head modified by hexose, glucuronic acid, inositol, and phosphate [43]. Three Sph molecular species were detected in wild-type and mutant ovules, including two PhytoSph and one Sph molecular species. In 0-DPA ovules, the content of Sph t18:1 was the highest, followed by Sph t18:0 and Sph d18:1 (Figure 4A) and the contents of Sph t18:0 and Sph d18:1 were decreased in mutants when compared with wild-type, among which, the reduction of Sph t18:0 in the *Xufl* mutant and Sph d18:1 in the *Xinfl* mutant was significant (Figure 4A). There are two molecular species of S1P, S1P t18:1 and S1P t18:0, which were detected in 0-DPA ovules, indicating that only trihydroxyl LCB was phosphorylated in 0-DPA ovules. The content of S1P t18:1 was higher than that of S1P t18:0 (Figure 4A). Twelve molecular species of Cer were detected in 0-DPA ovules. Among which, the content of Cer containing unsaturated LCB was higher while the Cer containing saturated LCB was lower (Figure 4B). Compared with wild type, the contents of Cer d18:1/22:0 and Cer d18:1/24:0 increased obviously in the two mutants excepted the level of Cer d18:1/24:0 in the *Xufl* mutant. Among them, the Cer d18:1/22:0 was increased by 2 fold and Cer d18:1/24:0 was increased by 23.6~46%. Compared with wild type, the Cer d18:0/22:0 also increased significantly in the two mutants, even though the Cer d18:0/22:0 had lower concentration in 0-DPA ovules. (Figure 4B). A total of 30 PhytoCer molecular species were detected in 0-DPA ovules, including 13 PhytoCers and 17 PhytoCer-OHFA. Two PhytoCer molecular species containing saturated C22 and C24 FA were predominant in 0-DPA ovule. Compared with wild type, the contents of PhytoCer t18:0/22:0 and PhytoCer t18:0/h22:1 were significantly increased in the two mutants (Figure 4C). These results showed that the Cer molecular species containing C22 and C24 FA were increased significantly in 0-DPA ovules of the two mutants.

### 2.4. The Difference of Complex Sphingolipids between Wild-Type and Mutants

Complex sphingolipids include GluCer and GIPC. A total of 17 GluCer molecular species were detected in 0-DPA ovules, including 8 Phyto-GluCers and 9 GluCers. The contents of two Phyto-GluCers, t18:1/h22:0 and t18:1/h24:0 were much higher than that of other GluCer molecular species (Figure 5A). In general, the contents of Phyto-GluCer were higher than that of GluCer. Among Phyto-GluCer molecular species, the molecule containing VLCFA was highly enriched in 0-DPA ovules, while the molecule containing LCFA was very low (Figure 5A). On the contrary, the content of GluCer is low in 0-DPA ovules, especially the molecule containing VLCFA (Figure 5A). All GluCer and Phyto-GluCer molecules contained desaturated LCB and hydroxylated saturated FA except two molecules, GluCer d18:1/h24:1 and d18:0/h20:0 (very low content) (Figure 5A). Compared with wild type, the contents of all GluCer molecular species were slightly decreased in the two mutants but not significantly. Two GIPC molecular species were detected in 0-DPA ovules, t18:1/h22:0 and t18:1/h24:0, which contain trihydroxyl desaturated LCB and hydroxylated saturated VLCFA. The content of GIPC t18:1/h24:0 is slightly higher than that of GIPC t18:1/h22:0. Compared with the wild type, the content of both GIPC were decreased in the two mutants although there was on significant difference between wild type and mutants (Figure 5B). These results indicated that the contents of GluCer and GIPC may be declined in the ovules of the two mutants.

### 2.5. The Expression Level of GhGCSs and GhIPCSs in Wild-Type and Mutant Ovules

According to the results of sphingolipids detection, the contents of Cer were increased, while GluCer and GIPC were decreased in the two mutants. It is speculated that the conversion from Cer to GluCer and GIPC is suppressed in the two mutants. The first step of GIPC synthesis is the transfer of phosphatidylinositol (PI) to ceramide, which was catalyzed by phosphatidylinositol ceramide synthase (IPCS), and then the addition of glucuronic acid to IPC by glycosyltransferase (IPUT1) [44]. GluCer synthase (GCS) catalyzes reaction from Cer to GluCer. There are 4 GCS and 8 IPCS annotated in the upland cotton genome (https://cottonfgd.org/) (accessed on 6 June 2021). We analyzed the expression level of *GhGCSs* and *GhIPCSs* in wild-type and mutant ovules. The results showed that the expression levels of all detected *GhIPCS* genes and three *GhGCS* genes were lower in the mutants than in the wild type (Figure 6). It is suggested that the synthesis of GluCer and GIPC was suppressed in the two *lintless-fuzzless* mutants, which was consistent with the results of content detection.

### 2.6. The Expression of Genes Related to Sterol and Sterol Ester Synthesis in the Two Mutant Ovules

Compared with the wild-type 0-DPA ovules, two sterols (campesterol and stigmasterol) and two steryl esters (campesterol esters and stigmasterol esters) were significantly increased in the ovules of the two mutants, while cholesterol was declined in the 0-DPA ovules of the two mutants. It is speculated that the synthesis and metabolism of sterol and steryl esters have been altered in mutants of 0-DPA ovules. Therefore, we analyzed the expression levels of genes involved in sterol and sterol ester synthesis in 0-DPA ovules of wild type and mutants. We selected the major enzymes of plant sterol biosynthesis, HMGR (3-hydroxy-3-methylglutaryl-CoA reductase), *SMT1* and *SMT2* (C24-sterol methyltransferases 1 and 2)and *CYP710A1* and *CYP710A2* (C22-sterol desaturase) [45]. PSAT (phospholipid:sterol acyltransferase) is involved in steryl esters synthesis in plant [46,47]. CPI1-1 (cyclopropylsterol isomerase1-1) is required for PIN protein localization [34]. *GhDWF1* is homologue of SSR2 (sterol side chain reductase 2) that catalyzes cycloartenol to cycloartanol [48]. The results showed that the expression levels of *GhHMGR1*, *GhDWF1*, and *GhCPI1* gene were decreased in the two mutants, while the expression levels of *GhCYP710A1*, *GhCYP710A2*, and *GhPASAT1* genes were increased in the mutants when compared with the wild type (Figure 7). This indicated that sterol and sterol ester synthesis was disrupted in the two mutants.

### 2.7. Exogenous Application of PDMP Inhibited Fiber Cell Initiation and Elongation

PDMP (1-phenyl-2-decanoylamino-3-morpholino-1-propanol) is a specific inhibitor of glucosylceramide synthase (GCS) [43]. In order to verify the role of GluCer in the initiation of fiber cells, we applied PDMP to in vitro culture system. After five-day culture, we could found fiber cells on the ovule surface in the mock treatment (Figure 8A), while there was almost no fiber cell on the surface of ovule treated by PDMP (Figure 8B). Fiber initiation and growth were further observed by SEM. It was found that the fiber cells on mock ovule were very long (Figure 8C), while only few short fiber cells were observed on the surface of ovule treated with PDMP, and the morphology of the treated fiber cells was abnormal (Figure 8D). The results showed that PDMP treatment seriously inhibited the initiation and elongation of fiber cells, suggesting that GluCer played some roles in the initiation and elongation of fiber cells.

## 3. Discussion

Phytosterols have structural and functional roles in plant growth and development [49,50], and its role in numerous physiological processes is well established, such as cell division, cell elongation, vascular element development, and tolerance to abiotic and/or biotic stress [50,51,52]. Phytosterols also play vital roles in ovule growth and fiber elongation in cotton [41,53,54,55]. However, the role of sterols in the differentiation and initiation of fiber cell is still unclear. On one hand, campesterol, as a precursor of the important plant hormone brassinosteroids (BRs), may influence the synthesis of BRs [49]. Furthermore, it has been reported that sterols were bioactive molecules independent BRs [28]. On the other hand, sterols, act as crucial components of lipid raft, regulate the permeability and fluidity of membrane, and the activity of membrane binding protein. The reported proteins that play an important part in the development of fibers are closely related to membrane function [56]. In this study, the composition and content of sterols changed in the 0-DPA ovules of two *lintless-fuzzless mutants* were compared with that in 0-DPA ovules of upland cotton wild type, which might be related to their fiber cell differentiation and initiation failure phenotype.

In *Arabidopsis thaliana* sterol synthesis mutants such as smt1 (sterol C-24 methyltranferase 1) and cpi1 (cyclopropylsterol reductase), the alteration of sterol composition and content resulted in the abnormal subcellular localization of PIN proteins, which in turn, disrupts the polar transport of auxin [33,34,57], and then led to a severe dwarf phenotype and a strong defect in the root gravitropism response in the *cpi* mutant, and a short root, small leaf, and multiple primary inflorescences phenotype in the smt1 mutant. Auxin is a very important plant hormone, which plays crucial roles in many aspects of plant growth and development. Auxin also plays a crucial role in the differentiation and initiation of cotton fiber cells [58,59]. Ectopic expression of *iaaM* gene in ovule epidermis of transgenic cotton could promote the initiation number of fiber cells [60]. In the present study, the expression level of *GhSMT1* gene is similar between the wild-type and two mutants. However, the expression level of *GhCPI1* gene significantly declined in the 0-DPA ovules of two mutants which might act similarly as cpi mutant and the auxin polar transport was disrupted in these two mutants, and at last, resulting in no fiber cell protruded from the epidermis of ovules. The relevant study about this hypothesis is in progress.

Previous study indicated that the balance between different sterols is one of the important modes for the functions of sterols. Both high sitosterol and low campesterol (*SMT2* overexpressors) and low sitosterol and high campesterol (*SMT2* suppression plants) seriously affected the growth and development of plants. It is considered that the ratio of campesterol to sitosterol (C/S) is an important factor affecting plant growth and development [40]. During the growth and development of cotton fiber cells, the ratio of C/S changed greatly in different developmental stages, such as elongation period (primary wall synthesis stage) present higher C/S value and secondary wall synthesis stage present lower C/S value [41,53]. In the study, the content of sitosterol has no significant difference between the two mutants and wild-type, while the content of campesterol was significantly higher in the two mutants, resulting in a significant increase in the C/S value (Figure 3B). It was to say that the balance between campesterol and sitosterol in the 0-DPA ovules of the two mutants was broken, which may be related to the inhibition of fiber cell initiation. Meanwhile, the balance between sitosterol and stigmasterol also plays an important role in plant resistance to biotic and abiotic stresses and plant growth. The ratio of stigmasterol to sitosterol is an important indicator of the state of plant membranes [45]. *AtCYP710A1* gene encoding C22-sterol desaturase, which catalyzes the reaction of β-sitosterol conversion into stigmasterol, plays a key role in resistance of *Arabidopsis thaliana* to low and high temperatures [61]. It has been reported that stigmasterol is a “stress” sterol. The induction of stigmasterol synthesis after receiving a pathogen is mediated by compounds such as flagellin, polysaccharides, and reactive oxygen species (ROS) [62,63]. Interestingly, it has been reported that the fiber initiation retardation in XinFLM might be related to the production of ROS [18,64]. In the present study, the content of stigmasterol in the 0-DPA ovules of the two mutants was significantly higher than that of wild type. Consistently, the expression levels of *GhCYP710A1* and *GhCYP710A2* genes were also significantly higher in the two mutants. This resulted in the ratio of sitosterol to stigmasterol changed dramatically in ovules of these two mutants. However, whether this alteration is the cause of mutant fiber initiation failure needs to be further studied. Future studies on the related mechanisms of regulating *GhCYP710A* will further clarify the functions of *GhCYP710A* and its products in fiber differentiation and protrusion.

The role of conjugated sterols in plant growth and development is unclear. A few reports showed that conjugated sterols are involved in many processes of plant growth and development. Conjugated sterols are ubiquitously found in plants but their relative contents highly differ among species and their profile may change in response to developmental and environmental cues. Sterol ester play a central role in membrane sterol homeostasis and also represent a storage pool of sterols in particular plant tissues [46].

In the present study, the contents of campesterol ester and stigmasterol ester in 0-DPA ovules of the two mutants were significantly increased. Simultaneously, the contents of free campesterol and stigmasterol also increased significantly, and the expression level of *GhPASAT1* gene, which is responsible for sterol ester synthesis in plant also increased. It is speculated that higher expression of *GhPASAT1* gene and higher sterol ester might rescue the abnormal concentration of free sterol. Plants strictly regulate the levels of sterol in their cells, as high sterol levels are toxic. HMGR is regard as a sterol sensor in plants. Much of the effort in understanding regulation of phytosterol biosynthesis has been focused on the role of HMGR. Evidence has been given for a good correlation between the level of HMGR activity and the rate of sterol production [65,66]. *GhHMGR1* was down-regulated in two mutant ovules maybe further suggest some sterol was overproduced. Consistently, the *HMGR1* and *HMGR2* were down-regulated in the high sterol ester 1 (hise1) mutant, in which abundantly accumulated sterol esters in leaf cells [67]. Therefore, the overproduction of campesterol and stigmasterol might be a key factor for the failure of fiber cell differentiation and initiation in the two mutants.

The content of cholesterol, as our result indicated, is very low in plants. However, it has also been reported that it plays a role in plant growth and development in recent years [48,68,69]. The smt1 mutant has an altered sterol content: it accumulates cholesterol and has less sitosterol content, and unaffected campesterol content. The smt1 plants have pleiotropic defects: poor growth and fertility, sensitivity of the root to calcium, and a loss of proper embryo morphogenesis [69]. In our study, the 0-DPA ovules of the two mutants had less cholesterol and more campesterol, and unaffected sitosterol content. Additionally, there was no obvious difference on *GhSMT1* expression level between wild type and the two mutants. It is suggested that other gene for cholesterol synthesis might be disrupted. Sonawane et al. had reported that the SSR2 (sterol side chain reductase 2) catalyzes the conversion from cycloartenol to cycloatanol, which is the first step of cholesterol biosynthesis pathway in plant [48]. In cotton genome, the homologue of SSR2 is *GhDWF1* [70]. We showed here that the expression level of *GhDWF1* gene strikingly declined in the two mutants. It is suggested that cholesterol might play a role in fiber cell initiation.

Sphingolipids are not only a crucial component of biomembranes but also are important bioactive molecules that mediates a variety of cell processes [43], such as programmed cell death [71,72,73,74], low temperature signal transduction [75,76,77], pathogen-induced hypersensitivity [72], host–pathogen crosstalk [78], the closure of stomatal guard cells regulated by ABA signaling [79,80,81,82], and the regulation of membrane stability [83]. Recent studies indicated that sphingolipids also participate in cotton ovule growth and fiber cell development [84,85]. Cer is an important intermediate of sphingolipid synthesis pathway, which can further generate GluCer and IPC (IPC further generates GIPC in Golgi). Compared with wild type, the content of Cer was increased, while the content of GluCer and GIPC was decreased, indicating that the flow from Cer to GluCer and GIPC was inhibited. Consistently, the expression levels of the genes responsible for GluCer and IPC synthesis was also significantly decreased in the two mutants. Furthermore, treatment of wild-type ovules with PDMP, a specific inhibitor of GCS, strongly inhibited the initiation of cotton fibers, indicating that the appropriate contents of Cer, GluCer and GIPC had some roles in the initiation of fiber cells. In future studies, we would like to reveal the mechanism underlying the functions of GCSs and IPCs in fiber initiation.

## 4. Materials and Methods

### 4.1. Plant Materials and Growth Conditions

The cotton plants used in this study were upland cotton (*Gossypium hirsutum* L.) cv. Xuzhou142 (XuFL), *Xuzhou142 lintless-fuzzless mutant* (*Xufl*) and *Xinxiangxiaoji lintless-fuzzless mutant* (*Xinfl*), which were provided by the Institute of Cotton Research, Chinese Academy of Agricultural Sciences and were grown under natural field conditions with normal administrations in Chongqing.

### 4.2. Sample Collection

The 0-DPA ovules (about 500 ovules in each sample) were collected from wild-type (XuFL), two *lintless-fuzzless mutants* (*Xufl* and *Xinfl*) at the day of anthesis, and immediately put into liquid nitrogen, and then kept at −80 °C.

### 4.3. Lipid Extraction and Lipidomics

After sample collection was completed, lipid extraction and lipidomic analysis were performed by the Lipidall Technologies Company Limited (http://www.lipidall.com/) (accessed on 16 August 2020), as described previously [27,85,86,87]. Briefly, the analyses were conducted using an Exion ultra-performance liquid chromatograph (UPLC) (AB Sciex, CA, USA) coupled with a Sciex QTRAP 6500 PLUS (AB Sciex, CA, USA). The lipids were separated using a Phenomenex Luna 3 µm silica column (Phenomenex, CA, USA) (internal diameter: 150 × 2.0 mm) under the following conditions: mobile phase A (chloroform: methanol: ammonium hydroxide, 89.5:10:0.5) and mobile phase B (chloroform: methanol: ammonium hydroxide: water, 55:39:0.5:5.5). The gradient began with 95% of mobile phase A for 5 min and was followed by a linear reduction to 60% mobile phase A over 7 min. The gradient was held for 4 min, and mobile phase A was then further reduced to 30% and was held for 15 min. MRM transitions were constructed for a comparative analysis of the various sphingolipids. The individual sphingolipid classes were quantified by referencing spiked internal standards, namely Cer d18:1/17:0, GluCer d18:1/12:0, d17:1-S1P, D-ribo-phytosphingosine C17, and d17:1-Sph from Avanti Polar Lipids (Alabaster, AL, USA) and GM1 d18:1/18:0-d3 from Matreya LLC. (State College, PA, USA).

Free sterols and steryl esters were analysed under atmospheric pressure chemical ionization (APCI) mode on a Jasper HPLC coupled to Sciex 4500 MD as described previously, using d6-cholesterol and d6-C18:0 cholesteryl ester (CE) (CDN isotopes) as internal standards [88].

### 4.4. RNA Extraction and qRT-PCR

Total RNA of 0-DPA ovules (about 100 ovules in each sample) from XuFL, *Xufl* and *Xinfl* was extracted using the RNAprep pure Plant Kit (TIANGEN, Beijing, China). First-strand cDNAs were synthesized using the PrimeScript™ RT reagent Kit with gDNA Eraser (TAKARA, Kyoto, Japan). qRT-PCR analysis was performed using Novostar-SYBR Supermix (Novoprotein, Shanghai, China): 94 °C for 2 min, followed by 40 cycles of 94 °C for 30 s, 56 °C for 30 s, and 72 °C for 1 min. Three biological repetitions were performed. The specific primers of selected genes and the internal control HISTONE3 (GenBank accession no. AF024716) are listed in Appendix A.

### 4.5. In Vitro Ovule Culture and Scanning Electron Microscopy

For the in vitro ovule cultures, cotton ovules (*Gossypium hirsutum* L.) were collected at the day of anthesis, sterilized in a 3‰ H_2_O_2_ solution, and cultured in Beasley and Ting’s (BT) medium [59] at 32 °C in the dark for five days. Meanwhile, for PDMP (1-phenyl-2-decanoylamino-3-morpholino-1-propanol) treatment assays, the ovules were cultured at 32 °C in darkness in BT medium with 60 μM PDMP. BT medium adjusted with the amount of DMSO (dimethyl sulfoxide) equivalent to that used to dissolve PDMP was used as mock. The cultured ovules (mock and PDMP treated ovules) were observed and took photographs by ALTO 1000E scanning electron microscope (SEM).

### 4.6. Statistical Data Analysis

Data were presented as mean ± SD. Statistical data analysis were performed by the one-tailed student’s *t*-test. * and ** indicate significant differences at *p* < 0.05 and *p* < 0.01, respectively.

## 5. Conclusions

Cotton fiber is a single cell seed trichome. Only about 30% of the seed epidermal cells differentiate into spinnable fibers in common upland cotton. Therefore, the timing and number of fiber initiation are closely correlated with the yield and quality of mature cotton fiber. However, the mechanism of fiber cell differentiation and initiation remains unclear. Here, we analyzed the difference of sphingolipids and sterols between wild type (XuFL) and two lintless-fuzzless mutants (*Xufl* and *Xinfl*) in 0-DPA (the day of anthesis) ovules. Compared with wild type, two Cer molecular species, d18:1/22:0 and d18:1/24:0 and two Phyto-Cer molecular species, t18:0/22:0 and t18:0/h22:1, were significantly increased, while the contents of all GluCer and GIPC molecular species were decreased. Consistently, the expression levels of seven genes involved in GluCer and GIPC synthesis decreased in the two mutants, suggesting the conversion from Cer to GluCer and GIPC were suppressed. Blocking the conversion by PDMP (1-phenyl-2-decanoylamino-3-morpholino-1-propanol), a specific inhibitor of GluCer synthase, significantly inhibited the initiation of cotton fiber cells. Meanwhile, the contents of stigmasterol and campesterol increased significantly, while cholesterol decreased in the two mutants. The content of total sterol esters increased significantly, mainly resulting from the content of campesterol esters, stigmasterol esters increased significantly in two mutants. Consistently, the expression levels of some sterol synthase genes and sterol ester synthase genes were altered in two mutants. These results revealed that sphingolipids and sterols have some roles in the initiation of fiber cell. Our study provides a novel insight into the regulatory mechanism of fiber cell initiation.

## Figures and Tables

**Figure 1 ijms-22-11438-f001:**
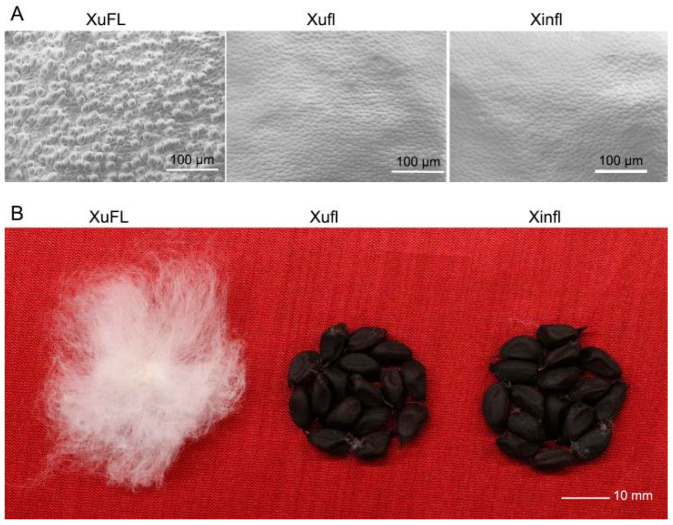
The phenotype of XuFL, *Xufl*, and *Xinfl*. (**A**) The scanning electron microscope images of 0-DPA ovules of XuFL, *Xufl*, and *Xinfl*. (**B**) The mature fibers and seeds of XuFL, *Xufl*, and *Xinfl*. XuFL, wild-type Xuzhou 142; *Xufl*, *Xuzhou 142 lintless-fuzzless mutant*; *Xinfl*, *Xinxiangxiaoji lintless-fuzzless mutant*.

**Figure 2 ijms-22-11438-f002:**
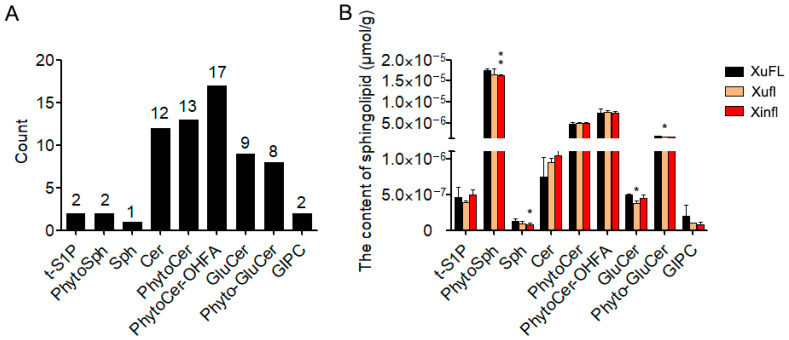
Sphingolipid classes in the 0-DPA ovules of cotton and their differences between wild-type and two mutants. (**A**) The number of classes and molecular species of sphingolipids detected in 0-DPA ovules. (**B**) The content of sphingolipid classes at the 0-DPA ovules of XuFL, *Xufl*, and *Xinfl*. Cer, ceramides; PhytoCer, phytoceramides; PhytoCer-OHFA, phytoceramides with hydroxylated fatty acyls; t-S1P, phytosphingosine-1-phosphate; Sph, sphingosines; PhytoSph, phytosphingosines; GluCer, glucosylceramides; Phyto-GluCer, phyto-glucosylceramides; GIPC, glycosyl-inositol-phospho-ceramides. XuFL, wild-type Xuzhou 142; *Xufl*, *Xuzhou 142 lintless-fuzzless* mutant; *Xinfl*, *Xinxiangxiaoji lintless-fuzzless* mutant. SD represents three biological repeats. Statistical data analysis was performed by the one-tailed student’s *t*-test. One asterisk and two asterisk indicate significant differences at *p* < 0.05 and *p* < 0.01, respectively.

**Figure 3 ijms-22-11438-f003:**
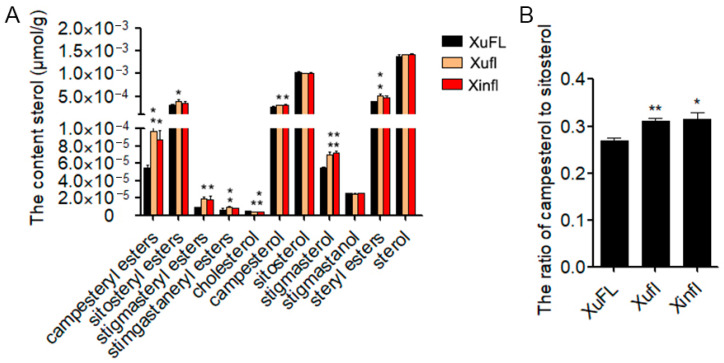
Sterol classes in the 0-DPA ovules of cotton and their differences between wildtype and two mutants. (**A**) The content of various sterol molecular species at the 0-DPA ovules of XuFL, *Xufl*, and *Xinfl*; (**B**) The ratio of campesterol to sitosterol. XuFL, wild-type Xuzhou 142; *Xufl*, *Xuzhou 142 lintless-fuzzless* mutant; *Xinfl*, *Xinxiangxiaoji lintless-fuzzless* mutant. SD represents three biological repeats. Statistical data analysis was performed by the one-tailed student’s *t*-test. One asterisk and two asterisk indicate significant differences at *p* < 0.05 and *p* < 0.01, respectively.

**Figure 4 ijms-22-11438-f004:**
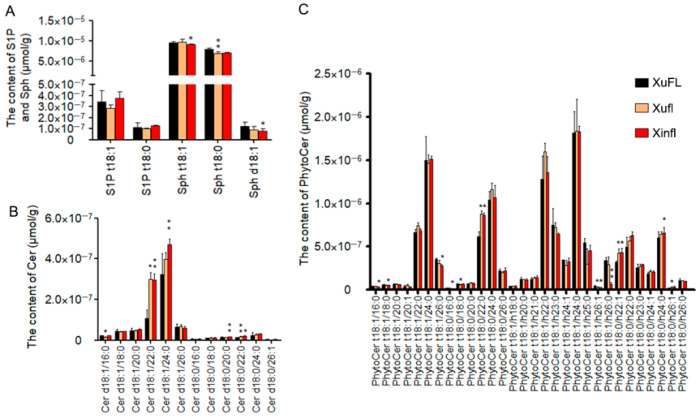
The difference of simple sphingolipid contents in the 0-DPA ovules between wild-type and two mutants. (**A**) The content of various molecular species of Sph and S1P. (**B**) The content of various molecular species of Cer. (**C**) The content of various molecular species of PhytoCer. Sph, sphingosines; S1P, sphingosine-1-phosphate; Cer, ceramides; PhytoCer, phytoceramides. “d18:0/1” indicates that the long-chain bases (LCB) of sphingolipids had two hydroxyl groups (d), 18 carbon atoms, and no or 1 double bond; “t18:0/1” indicates that the LCB had three hydroxyl groups (t), 18 carbon atoms, and no or 1 double bond; “16-26:0/1” indicates that the long-chain fatty acid (LCFA) or very long chain fatty acid (VLCFA) of sphingolipids had 16 to 26 carbon atoms and no or one double bond; and “h16-26:0/1” indicates that the long-chain fatty acid (LCFA) or very long chain fatty acid (VLCFA) of sphingolipids was a hydroxylated fatty acyl (h) and had 16 to 26 carbon atoms and no or 1 double bond. XuFL, wild-type Xuzhou 142; *Xufl*, *Xuzhou 142 lintless-fuzzless* mutant; *Xinfl*, *Xinxiangxiaoji lintless-fuzzless* mutant. SD represents three biological repeats. Statistical data analysis was performed by the one-tailed student’s t-test. One asterisk and two asterisk indicate significant differences at *p* < 0.05 and *p* < 0.01, respectively.

**Figure 5 ijms-22-11438-f005:**
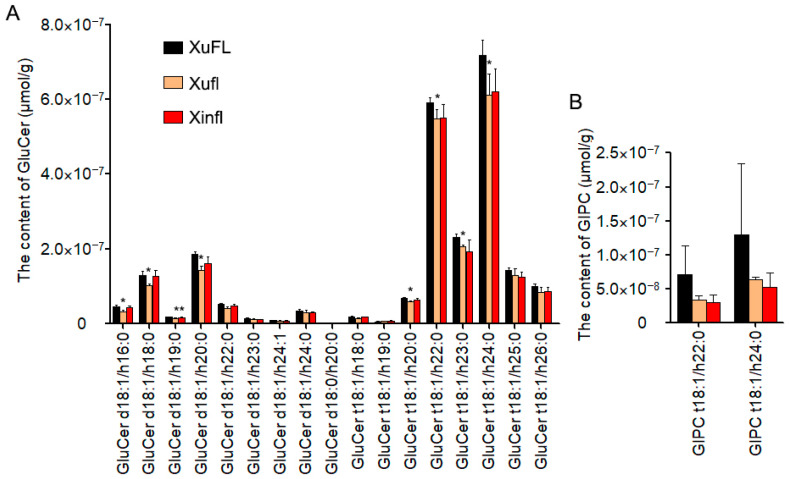
The difference of complex sphingolipids content in the 0-DPA ovules between wild-type and two mutants. (**A**) The content of various molecular species of GluCer. (**B**) The content of two molecular species of GIPC. GluCer, glucosylceramides; GIPC, glycosyl-inositol-phospho-ceramides. “d18:0/1” indicates that the long-chain bases (LCB) of sphingolipids had two hydroxyl groups (d), 18 carbon atoms and no or 1 double bond; “t18:0/1” indicates that the LCB had three hydroxyl groups (t), 18 carbon atoms, and no or 1 double bond; “16-26:0/1” indicates that the long chain fatty acid (LCFA) or the very long chain fatty acid (VLCFA) of sphingolipids had 16 to 26 carbon atoms and no or 1 double bond; and “h16-26:0/1” indicates that the long-chain fatty acids (LCFA) or the very long chain fatty acid (VLCFA) of sphingolipids were hydroxylated fatty acyls (h) and had 16 to 26 carbon atoms and no or 1 double bond. XuFL, wild-type Xuzhou 142; *Xufl*, *Xuzhou 142 lintless-fuzzless* mutant; *Xinfl*, *Xinxiangxiaoji lintless-fuzzless* mutant. SD represents three biological repeats. Statistical data analysis was performed by the one-tailed student’s *t*-test. One asterisk and two asterisk indicate significant differences at *p* < 0.05 and *p* < 0.01, respectively.

**Figure 6 ijms-22-11438-f006:**
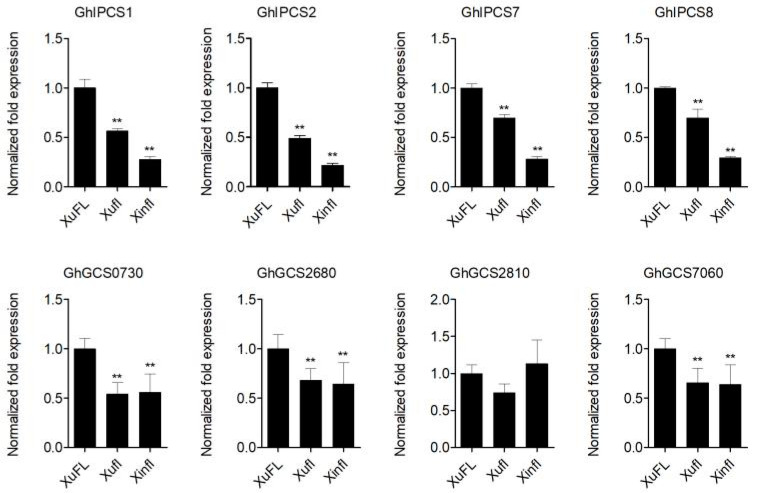
The differential gene expression for GluCer and IPC synthesis between wild-type and two mutants. *GhIPCS1*, *GhIPCS2*, *GhIPCS7*, and *GhIPCS8* are homologues of IPCS (phosphatidylinositol ceramide synthase) in cotton genome. *GhGCS0730*, *GhGCS2680*, *GhGCS2810*, and *GhGCS7060* are homologues of GCS (GluCer synthase) in cotton genome. XuFL, wild-type Xuzhou 142; *Xufl*, *Xuzhou 142 lintless-fuzzless* mutant; *Xinfl*, *Xinxiangxiaoji lintless-fuzzless* mutant. Three independent RNA isolations were used for cDNA synthesis, and each cDNA sample was subjected to quantitative real-time PCR analysis in triplicate. Error bars represent the SD. Statistical data analysis was performed by the one-tailed student’s *t*-test. ** indicate significant differences at *p* < 0.01.

**Figure 7 ijms-22-11438-f007:**
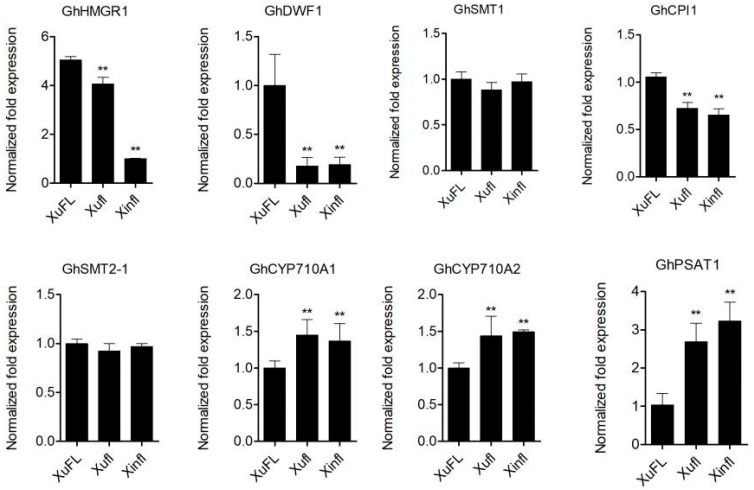
The differential gene expression for sterol and steryl ester synthesis between wild-type and two mutants. HMGR (3-hydroxy-3-methylglutaryl-CoA reductase), *SMT1* and *SMT2* (C24-sterol methyltransferases 1 and 2), and CYP710A1 and CYP710A2 (C22-sterol desaturase), PSAT (phospholipid:sterol acyltransferase), CPI1-1 (cyclopropylsterol isomerase1-1), *GhDWF1* (SSR2, sterol side chain reductase 2). XuFL, wild-type Xuzhou 142; *Xufl*, *Xuzhou 142 lintless-fuzzless* mutant; *Xinfl*, *Xinxiangxiaoji lintless-fuzzless* mutant. Three independent RNA isolations were used for cDNA synthesis, and each cDNA sample was subjected to quantitative real-time PCR analysis in triplicate. Error bars represent the SD. Statistical data analysis was performed by the one-tailed student’s *t*-test. ** indicate significant differences at *p* < 0.01.

**Figure 8 ijms-22-11438-f008:**
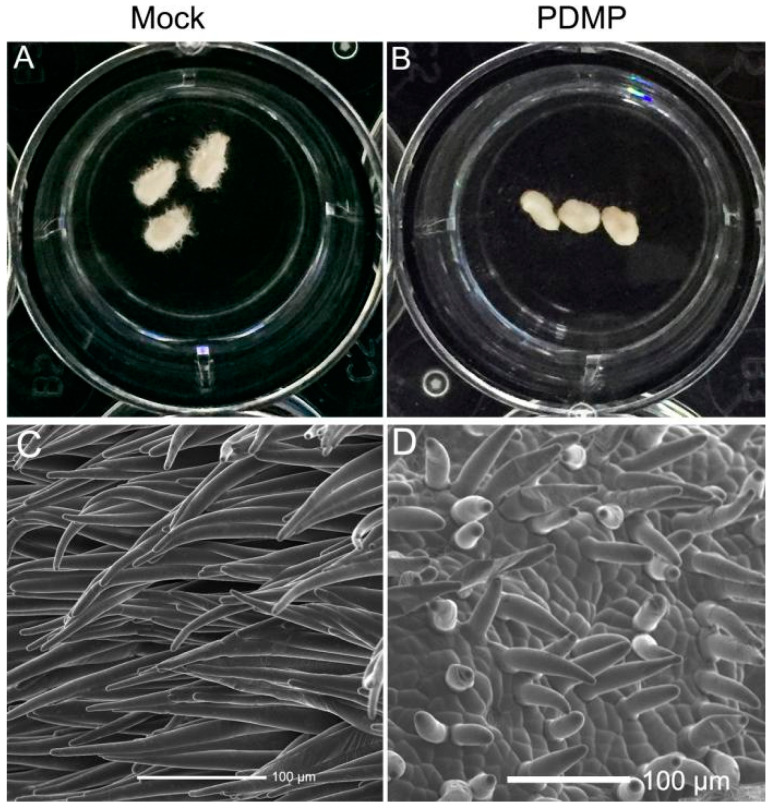
Exogenous application of PDMP in in vitro ovule culture system. (**A**) 0-DPA ovules of XuFL (wild-type) were cultured for five days in the medium adjusted with the amount of DMSO equivalent to that used to dissolve PDMP (Mock). (**B**) 0-DPA ovules of XuFL (wild-type) were cultured for five days in the medium with 60 μM PDMP. (**C**) The fiber cells on the ovules without treatment. (**D**) The fiber cells on the ovules treated with PDMP. PDMP, 1-phenyl-2-decanoylamino-3-morpholino-1-propanol.

## Data Availability

Not applicable.

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
