# Peer review of "Comparative Metabolomics Analysis Reveals Sterols and Sphingolipids Play a Role in Cotton Fiber Cell Initiation"

_ijms, 2021, doi:10.3390/ijms222111438_

Round 1

Reviewer 1 Report

Thank you for such an interesting work Only minor comments:

1)Please use abbreviations of the journals: The Plant Journal, New Phytologist, the Plant cell, Plant physiology, Biochemical and Biophysical Research Communications, Frontiers in Plants science, etc2) Use italics for cotton in ref 64,16,18, 8

Author Response

1.Please use abbreviations of the journals: The Plant Journal, New Phytologist, the Plant cell, Plant physiology, Biochemical and Biophysical Research Communications, Frontiers in Plants science, etc.

Answer:Thank you very much for your advice, we have modified the format according to your requirements.

2.Use italics for cotton in ref 64,16,18, 8.

Answer:Thank you for your suggestion, we have changed these words into the correct form.

Reviewer 2 Report

In this manuscript, authors analyzed the contents of sphingolipids and sterols in cotton ovules to elucidate the role of sphingolipids and sterols in cotton fiber cell development. They utilized cotton Gossypium hirsutum cv. Xuzhou142 (XuFL) as wild type and Xuzhou142 lintless-fuzzless (Xufl) and Xingxiangxiaoji lintless-fuzzless (Xinfl) mutants, in which fiber cell development is suppressed. Authors subjected 0 day post anthesis ovules to UHPLC-MS/MS, and found that many sphingolipid and sterol species are differently accumulated in mutants compared to wild type. Authors also analyzed expression levels of genes involved in biosynthesis of those sphingolipid/sterol species, and the expression changes of many genes were correlated with the changes in sphingolipid/sterol amount. Finally, authors treated cotton ovule culture with PDMP, which is a specific inhibitor for GluCer synthesis, and found that fiber cell growth was significantly suppressed. Based on those results, authors concluded that sphingolipid/sterol may play a role in cotton fiber cell growth.

One serious problem of this manuscript is a complete lack of information regarding the mutants utilized for analyses, as described below. Without this information, it is not possible to judge the scientific appropriateness of this study. Other minor issues that must be revised are listed below.

Major comment

  1. The information of mutants utilized for analyses is not sufficient. Have those two mutants been reported previously? If those mutants have been reported previously, authors must describe what is known for the mutants (how they were isolated, what are the phenotypes other than fiber development, and what are the causative genes) with adequate references. If those mutants have not been reported previously, authors provide more information regarding how they were isolated and other phenotypes. In addition, it is not clear the relationship between two mutants. Based on their names, I assume that Xufl and Xinfl both have the same mutation in the same gene but have different genetic backgrounds. Xufl seems to have the same genetic background with wildtype Xuzhou142, while the background of Xinfl seems to be different. If my assumption is correct, this manuscript contains serious problem because it is inappropriate to compare plants with different genetic backgrounds. Authors must give sufficient information regarding the relationship and genetic backgrounds of those two mutants. If wt and Xinfl have different genetic backgrounds and if there are any logical reasons for comparing plants with different backgrounds, describe the reasons. If wt and Xinfl have different genetic backgrounds and if there is no logical reason for comparing those two, I recommend to delete Xinfl

Minor comments

  1. This manuscript contains too many typographs. In Arabidopsis thaliana, gene name must be written in uppercase letter in italic, and mutant names must be written in lowercase letter in italic.
  2. “Arabidopsis” is a genus name, and “Arabidopsis” and “Arabidopsis thaliana” are different. The species name must be written correctly.
  3. In p4 L167, please define which are “complex” and “simple” sphingolipids.
  4. Figure 3A is a meaningless graph. Please delete.
  5. In this manuscript, authors compared the amount of different molecular species of sphingolipids and sterols between wt and mutants. While authors often described “XX was decreased (or increased) in two mutants”, this is not correct. For example, in p6 L229, authors described “Sph d18:1 were decreased in mutants”. However, Sph d18:1 was decreased only in Xinfl but not in Xufl according to Figure 4A.
  6. While authors described “all GluCer molecular species were decreased in the two mutants” (p7 L272) and “the content of both GIPC were decreased in the two mutants” (p276), Figures 5A and 5B indicates that the amounts of many GluCer species and both GIPC are not statistically different from wt and mutants. Authors must explain this discrepancy.
  7. For materials and methods, please describe approximately how many ovules are used for lipid analysis and qRT-PCR, respectively.
  8. Table S2 is not mentioned in the manuscript. All the figures and tables (including supplement) must be mentioned in the manuscript.

Author Response

Major comment:

1.The information of mutants utilized for analyses is not sufficient. Have those two mutants been reported previously? If those mutants have been reported previously, authors must describe what is known for the mutants (how they were isolated, what are the phenotypes other than fiber development, and what are the causative genes) with adequate references. If those mutants have not been reported previously, authors provide more information regarding how they were isolated and other phenotypes. In addition, it is not clear the relationship between two mutants. Based on their names, I assume that Xufl and Xinfl both have the same mutation in the same gene but have different genetic backgrounds. Xufl seems to have the same genetic background with wildtype Xuzhou142, while the background of Xinfl seems to be different. If my assumption is correct, this manuscript contains serious problem because it is inappropriate to compare plants with different genetic backgrounds. Authors must give sufficient information regarding the relationship and genetic backgrounds of those two mutants. If WT and Xinfl have different genetic backgrounds and if there are any logical reasons for comparing plants with different backgrounds, describe the reasons. If WT and Xinfl have different genetic backgrounds and if there is no logical reason for comparing those two, I recommend to delete Xinfl.

Answer: Thank you for your valuable suggestions. Xuzhou 142 fuzzless-lintless mutant (Xufl) and Xinxiang Xiaoji fuzzless-lintless mutant (Xinfl) are spontaneous fiber initiation mutants, which have been widely used in previous studies. The Xufl mutant was isolated from the cotton cultivar G. hirsutum cv. Xuzhou142 (Zhang, 1991). The Xufl plants show no phenotypic difference from the wild type (XuFL), except that the Xufl seeds are glabrous. Genetic analysis indicated that the Xufl mutant was derived from a single recessive mutation from the wild type of Xuzhou142 (Zhang, 1991). However, the causative genes are not clear so far. In the last two decades, The XuFL and its fuzzless-lintless mutant (Xufl) were widely used in studying the early development of cotton fiber cell (Yu, 2000, Hu et al., 2018, Liu et al., 2012, Wang et al., 2010). The Xinfl mutant was found in the cotton field of Xiaoji Town, Xinxiang County, Henan Province, China, in 1991. Genetic analysis showed that its wild type may be Yumian 4#. However, the causative genes also are not clear so far. There was no phenotypic difference, except for fuzzless-lintless phenotype between the Xinfl plants and cotton cultivar G. hirsutum cv. Xuzhou142 or Yumian 4#. Considering the very narrow genetic background among cultivars of upland cotton and there was no certain wild type of the Xinfl mutant, comparative analysis is often carried out between normal upland cotton cultivars such as XuFL and the Xinfl mutant (Tang et al., 2014). Moreover, Wang et al. reported that the index of genetic identity was 0.9 between the Xinfl mutant and the xufl mutant indicating the two fuzzless-lintless mutants had great genetic similarity (Wang, 2005). Therefore, the two mutants often used in the study of revealing the genes and regulatory mechanism related to cotton fiber cell initiation. Tang et al. reported there was no fiber cell initiation from the 0 DPA ovules of the Xufl mutant and the Xinfl mutant, and the expression level of GhCaM7 in 0 DPA ovules was lower in two fuzzless/lintless mutants than in the normal upland cotton lines Xuzhou 142 (Tang et al., 2014). In this study, to further confirm the relationship of different sphingolipids or sterols with fuzzless-lintless phenotype, we used the two mutants with same fuzzless-lintless phenotype and high genetic identity.

Reference:

HU, H., WANG, M., DING, Y., ZHU, S., ZHAO, G., TU, L. & ZHANG, X. 2018. Transcriptomic repertoires depict the initiation of lint and fuzz fibres in cotton (Gossypium hirsutum L.). Plant Biotechnol J, 16, 1002-1012.

LIU, K., HAN, M., ZHANG, C., YAO, L., SUN, J. & ZHANG, T. 2012. Comparative proteomic analysis reveals the mechanisms governing cotton fiber differentiation and initiation. Journal of Proteomics, 75, 845-856.

TANG, W. X., TU, L. L., YANG, X. Y., TAN, J. F., DENG, F. L., HAO, J., GUO, K., LINDSEY, K. & ZHANG, X. L. 2014. The calcium sensor GhCaM7 promotes cotton fiber elongation by modulating reactive oxygen species (ROS) production. New Phytologist, 202, 509-520.

WANG, Q. Q., LIU, F., CHEN, X. S., MA, X. J., ZENG, H. Q. & YANG, Z. M. 2010. Transcriptome profiling of early developing cotton fiber by deep-sequencing reveals significantly differential expression of genes in a fuzzless/lintless mutant. Genomics, 96, 369-376.

WANG, S. H., DU, X. M. 2005. SSR Fingerprinting Analysis on Distinct Mutants of Fiber Development in Gossypium hisutum. Scientia Agricultural Sinica, 38, 2139-2146.

YU, X., ZHU, Y., LU, S., ZHANG, T., CHEN, X., & XU, Z. 2000. A comparative analysis ofafuzzless-lintless mutant ofGossypium hirsutum L. cv. Xu-142. Science in China Series C: Life Sciences, 43, 623-630.

ZHANG, T. Z., PAN, J. J., 1991. Genetic analysis of fuzzless-lintless mutant in upland cotton. Jiangsu J. Agr. Sci., 7, 13-16.

Minor comments in order of appearance:

1.This manuscript contains too many typographs. In Arabidopsis thaliana, gene name must be written in uppercase letter in italic, and mutant names must be written in lowercase letter in italic.

Answer:Thank you very much for your suggestion, we accepted and corrected.

2.“Arabidopsis” is a genus name, and “Arabidopsis” and “Arabidopsis thaliana” are different. The species name must be written correctly.

Answer:Thank you for your suggestion, we have changed these word into the correct form.

3.In p4 L167, please define which are “complex” and “simple” sphingolipids.

Answer:Thank you for your advice. According to the previous review by Goñi and Alonso, Simple sphingolipids mainly include four molecular species, namely sphingosine (Sph), ceramides (Cer) or phytoceramides, and their phosphorylated derivatives sphingosine-1-phosphate (S1P) and ceramide-1-phosphate (CerP). Complex sphingolipids include two molecular species, GCS and GIPC. In opposition to simple sphingolipids, complex sphingolipids have complex head modified by hexose, glucuronic acid, inositol, and phosphate.

4.Figure 3A is a meaningless graph. Please delete.

Answer:Thank you for your suggestion, we have deleted the figure 3A from the paper.

5.In this manuscript, authors compared the amount of different molecular species of sphingolipids and sterols between wt and mutants. While authors often described “XX was decreased (or increased) in two mutants”, this is not correct. For example, in p6 L229, authors described “Sph d18:1 were decreased in mutants”. However, Sph d18:1 was decreased only in Xinfl but not in Xufl according to Figure 4A.

Answer:Thank you for your advice. We examined and revised the relevant description to clearly describe whether the increase and decrease in quantity reached a significant level.

6.While authors described “all GluCer molecular species were decreased in the two mutants” (p7 L272) and “the content of both GIPC were decreased in the two mutants” (p276), Figures 5A and 5B indicates that the amounts of many GluCer species and both GIPC are not statistically different from wt and mutants. Authors must explain this discrepancy.

Answer:Thank you for your advice. We examined and revised the relevant description to clearly describe whether the increase and decrease in quantity reached a significant level.

7.For materials and methods, please describe approximately how many ovules are used for lipid analysis and qRT-PCR, respectively.

Answer:Thank you for your suggestion, we have accepted and added.

8.Table S2 is not mentioned in the manuscript. All the figures and tables (including supplement) must be mentioned in the manuscript.

Answer:Thank you for your suggestion, we have accepted and added.